# In situ imaging reveals disparity between prostaglandin localization and abundance of prostaglandin synthases

Kyle D. Duncan [1,5], Xiaofei Sun [2,3,5], Erin S. Baker [4], Sudhansu K. Dey [2,3] & Ingela Lanekoff [1✉]

Prostaglandins are important lipids involved in mediating many physiological processes, such as allergic responses, inflammation, and pregnancy. However, technical limitations of in-situ prostaglandin detection in tissue have led researchers to infer prostaglandin tissue distributions from localization of regulatory synthases, such as COX1 and COX2. Herein, we apply a novel mass spectrometry imaging method for direct in situ tissue localization of prostaglandins, and combine it with techniques for protein expression and RNA localization. We report that prostaglandin $D_2$, its precursors, and downstream synthases co-localize with the highest expression of COX1, and not COX2. Further, we study tissue with a conditional deletion of transformation-related protein 53 where pregnancy success is low and confirm that PG levels are altered, although localization is conserved. Our studies reveal that the abundance of COX and prostaglandin $D_2$ synthases in cellular regions does not mirror the regional abundance of prostaglandins. Thus, we deduce that prostaglandins tissue localization and abundance may not be inferred by COX or prostaglandin synthases in uterine tissue, and must be resolved by an in situ prostaglandin imaging.

[1] Department of Chemistry-BMC, Uppsala University, Uppsala, Sweden. [2] Division of Reproductive Sciences, Cincinnati Children's Hospital Medical Center, Cincinnati, OH 45229, USA. [3] College of Medicine, University of Cincinnati, Cincinnati, OH 45221, USA. [4] Department of Chemistry, North Carolina State University, Raleigh, NC, USA. [5] These authors contributed equally: Kyle D. Duncan, Xiaofei Sun. ✉email: Ingela.Lanekoff@kemi.uu.se

Prostaglandins (PGs) are potent bioactive lipid mediators that play important roles in various biological processes, including angiogenesis, inflammation, and reproductive events[1,2]. PG biosynthesis is mediated by several key enzymes, including phospholipase A2 and diacylglycerol lipase, which cleave acylglycerol chains such as arachidonic acid from phospholipids and diacylglycerols. Subsequently, cyclooxygenases (COX1 and COX2) mediate the conversion of arachidonic acid to $PGH_2$, which is considered the rate-limiting step in PG biosynthesis (Fig. 1A)[3–5]. Of the two PG synthase enzymes, COX1 (encoded by Ptgs1) is largely considered as a basal PG regulator, whereas COX2 (encoded by Ptgs2) is considered an inducible enzyme[6,7]. After $PGH_2$ formation, distinct downstream PG synthases engage in synthesizing specific PGs, such as $PGD_2$, $PGE_2$, and $PGF_{2\alpha}$ (Fig. 1A). Although the molecular structures of PGs are similar, individual PGs have different biological functions.

In pregnancy, PGs are key regulatory molecules in multiple events, including ovulation, fertilization, embryo implantation, decidualization, and parturition[8]. Two major regions in embryo implantation sites are the mesometrial (M) pole and the antimesometrial (AM) pole (Fig. 1B, C, and D). Embryo homing and implantation of the developing embryo are presumed to occur at the AM pole, which decidua is the main provider of nutrition prior to the establishment of a functional placenta at the M pole[9]. The importance of PGs in pregnancy is revealed by defective ovulation and fertilization failure in mice with a deficiency in Ptgs2[10,11]. To underscore the importance of COX2 in implantation, we have previously shown that $Ptgs2^{-/-}$ mice develop implantation failures, and further studies suggested that COX2-derived $PGI_2$ plays a key role in embryo implantation[12]. Mice deficient in Ptgs1 show no apparent implantation defects[13], but a compensatory expression of Ptgs2 is observed for Ptgs1 in a similar pattern as Ptgs2[14], which suggests that PGs derived from COX1 play a critical role during implantation. As implantation and decidualization progress, Ptgs2 expression is switched from the antimesometrial (AM) pole to the mesometrial (M) pole of the implantation chamber[11]. On day 8 of pregnancy, the decidua is fully differentiated into primary and secondary decidual zones, and both COX1 and COX2 are expressed. By day 10, COX2 is expressed around the fetal-maternal interface around the ecto-placental cone, the presumptive site of placentation[15], and by midgestational stage, COX2 is observed in invasive spiral-artery trophoblast giant cells[16]. These results imply a role for PGs in placentation, tissue remodeling, and embryonic development. $PGF_{2\alpha}$ derived from COX1 is considered indispensable for on-time parturition and birth[17].

Previous studies have shown that transformation-related protein 53 (p53) plays a critical role in many pathophysiological processes, including pregnancy and embryonic development[18,19]. For example, uterine-specific deletion of p53 has been shown to promote preterm labor in mice and has been implicated in premature births in humans[18]. Furthermore, studies have revealed that p53 interacts with AMPK and mTORC1 signaling, which control parturition timing, and we have reported on specific lipid accumulation and depletion[19–21]. Further, evidence linking deletion of p53 to aberrations in COX2-derived PG signaling includes elevated levels of COX2 activity on day 16 of pregnancy[18]. The sum of these studies suggests a significant alteration in PGs also at the earlier day 8 of pregnancy. Considering the importance of p53 in healthy embryonic development and parturition, relatively little is known of p53's impact on specific PG synthases, PG receptors, PG precursors, or PG spatial distributions. In sum, PGs play a key role throughout the process of pregnancy. However, our knowledge of localization of specific PGs in pregnancy events remains limited due to the dynamic spatiotemporal pattern of PG synthetic enzymes and the measurable accumulation of unstable PGs in uteri at different stages and conditions of pregnancy.

Despite the biological importance of PGs, localization of native PGs to specific cells in tissues has not yet been possible, although methods using PGs labelled with biotins have helped in this regard[22,23]. As a result, PG distribution in tissues has been largely assumed to correspond to the regional expression of COX2 and/or downstream PG synthases. Mass spectrometry imaging (MSI)

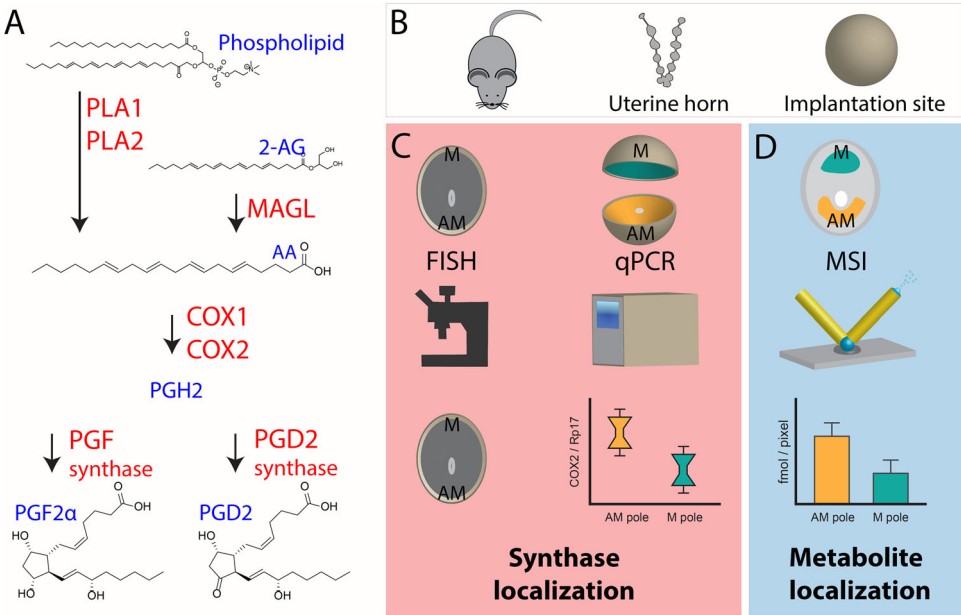

**Fig. 1 Schematic detailing the studied PG biosynthesis pathway and experimental design. A** Generation of PG species from phospholipids or 2-AG via COX and PG synthases, enzymes are in red and metabolites in blue. **B** Mouse embryo implantation sites from day 8 of pregnancy were used from $Trp53^{f/f}$ and $Trp53^{d/d}$ mice. **C** Florescence in situ hybridization was applied to study the in situ localization of synthases. Implantation sites were segmented into the AM-pole and the M-pole for RT-qPCR analysis of enzyme mRNA expression. **D** Quantitative nano-DESI MSI was performed on thin sections of implantation sites to determine in situ small molecule localization.

techniques have the ability to localize a wide range of molecules in thin tissue sections. However, although PG precursors, such as 2-arachidonoylglycerol (2-AG), arachidonic acid, and other arachidonic acid-containing phospholipids, have been imaged in tissues with MSI[20,24], localizing PGs to specific cellular regions has been a major technical challenge. This challenge largely arises from low PG ionization efficiencies in combination with very low endogenous concentrations, ranging between picomoles and nanomoles per gram of tissue[18,25–27]. Recently, we reported a novel approach using silver-doped nanospray desorption electrospray ionization (nano-DESI) MSI for quantitative imaging of prostaglandins in thin tissue sections of mouse implantation sites of day 4 of pregnancy[28]. In particular, within one experiment we uncovered the localization and quantity of four different PGs to the luminal epithelium and the glandular epithelium of the tissue, showing applicability for biological studies.

Here, we use the unique combination of silver doped nano-DESI MSI, RNA in situ hybridization, and RT-qPCR to study the correlation between enzyme expression and PG distribution in mouse embryo implantation sites on Day 8 of pregnancy. We apply this workflow to study the impact of p53 on the molecular machinery of PG biosynthesis in early embryonic development. The goal of this study was to: (i) visualize detectable PG species; (ii) detect any perturbations to PG synthesis as a result of p53 deletion; and (iii) correlate spatial distributions of PG species and PG synthases. Overall, this study provides the first evidence of PG localization in a tissue linked to a spatially defined analysis of key enzymes involved in PG biosynthetic pathways and demonstrates that the abundance or spatial distribution of PGs in tissue cannot be determined by monitoring only PG synthases.

## Results

The biosynthetic pathway for PGs includes several key enzymes (Fig. 1A), that were monitored in the workflow for spatial correlation between PG synthases, PG receptors, PG dehydrogenase, PG precursors, and PGs was established for mouse embryo implantation sites on day 8 of pregnancy (Fig. 1B−D). Specifically, the uterine horn was extracted from control mice and mice with a uterine conditional deletion of p53 on day 8 of pregnancy and individual embryo implantation sites were isolated (Fig. 1B). Expression of key synthases and dehydrogenase involved in PG biosynthesis and signaling pathways were analyzed using RT-qPCR. Specifically, dissected mouse uterine implantation sites were partitioned into AM and M-pole segments, and the mRNA expression was quantified by RT-qPCR for each pole of the implantation site (Fig. 1C and Table S1). As a result, the levels of mRNA for PG synthases could be statistically compared between the two morphologically different poles to explore regional PG biosynthesis on day 8 of pregnancy. To provide additional spatial information on PG biosynthetic enzyme localization, fluorescence in situ hybridization and immunostaining were performed on thin tissue sections of implantation sites (Fig. 1C).

Quantitative imaging of PGs and PG molecular precursors in implantation sites was performed by silver doped nano-DESI MSI with subsequent region-of-interest data analysis from molecules in the AM and M-pole (Fig. 1D). The analysis was performed untargeted to enable detection of all PGs and PG precursors in one experiment per tissue section, through localized liquid extraction of tissue molecules into a liquid bridge between two capillaries with subsequent electrospray ionization (nano-DESI, Fig. 1D)[20,28–30]. Despite the untargeted approach, only the two PG species PGD$_2$ and PGF$_{2\alpha}$ were detected at maximum 4 fmoles/pixel and 1.2 fmoles/pixel in control mice, respectively (Fig. 2A and S1). It is likely that there are additional PG species of importance present in the tissue; however, these were below the

limit of detection and therefore not included. Quantitative prostaglandin imaging of PGD$_2$ and PGF$_{2\alpha}$ was accomplished by introducing heavy isotope-labeled PG standards for direct single-point calibration in each pixel[28,30]. Isomeric identification determined that the detected isomer was PGD$_2$ and not PGE$_2$, which was accomplished by liquid chromatography ion mobility spectrometry and mass spectrometry (LC-IMS-MS) (Fig. S2 and Tables S2, S3). The amount of PG in morphological regions of the tissue was determined by extracting data from the AM pole and the M pole for all biological and technical replicate images to ensure accurate biological interpretations and correlation to regional expression of proteins followed by statistical analyses. Overall, our established workflow-enabled speciation, quantification, and localization of PGs, PG precursors, and expression of PG synthases.

**Regional distribution of PG and PG precursors**. Quantitative nano-DESI MSI enabled direct in situ imaging of PGD$_2$ (Fig. 2A) and PGF$_{2\alpha}$ (Fig. S1) in tissue from day 8 implantation sites. Simultaneously, the technique provided in situ imaging of the PG precursor's 2-AG and arachidonic acid (Fig. 2A and Table S4). The ion images presented in Fig. 2A are quantitative through the use of internal standards and the color scale of the pixels are individually represented for each ion image in fmoles/pixel. The results illustrate that PGD$_2$ and all its precursors localize to the AM-pole (Fig. 2A, B), while PGF$_{2\alpha}$ mainly localizes to the myometrium on day 8 of pregnancy (Fig. S1). Relative comparisons of the spatial distribution for PGD$_2$ and its precursors show that arachidonic acid is ~2.5 times higher in the AM-pole compared to the M-pole while 2-AG and PGD$_2$ are ~1.5 times higher in the AM-pole (Fig. 2B and Tables S5, S6).

Regional distribution of PG and PG precursors was investigated in mice with a uterine-specific deletion of Trp53. Quantitative nano-DESI MSI reveals significant accumulation of 2-AG, arachidonic acid, and PGD$_2$ in the AM-pole of the Trp53$^{d/d}$ mice (Fig. 2A, C). Compared to their detected abundances in the AM-pole of Trp53$^{f/f}$ mice, 2-AG is increased ~2 times, and arachidonic acid and PGD$_2$ ~1.5 times in Trp53$^{d/d}$ mice (Fig. 2C and Table S5, S6). Negligible differences were observed for PGF$_{2\alpha}$ in Trp53$^{f/f}$ and Trp53$^{d/d}$ mice, which were primarily detected in the myometrium (Figs. S1 and S3). Overall, changes in PG and PG precursor abundance as a result of p53 deletion appears to be isolated to the AM-pole, as no significant differences were observed for any of these molecules in the M-pole (Fig. 2A, C and Tables S5, S6).

**Regional distribution of PG synthases**. Fluorescence in situ hybridization (FISH) of COX2 protein shows synthase localization (Fig. 3A). Segmented RT-qPCR of mouse embryo implantation sites on day 8 of pregnancy further show that mRNA expression of Ptgs2 encoding COX2 is 15 times higher in the M-pole compared to the AM-pole (Fig. 3B). Results from RT-qPCR and FISH of Ptgs1 both reveal that Ptgs1 expression is primarily found in the AM-pole of the uterine implantation site (Fig. 3A, B). Expression of the two PGD$_2$ synthases Ptgds[31] and Hpgds[32] with segmented RT-qPCR shows that both are 2 and 4 times higher in the AM-pole than in the M-pole of the implantation sites on day 8 of pregnancy, respectively (Fig. 3A, B).

Regional distribution of PG synthases was studied in control mice and mice with a uterine-specific deletion of Trp53. Data from segmented RT-qPCR show that Ptgs2 expression is significantly higher in the AM-pole of p53 deficient mice compared to control mice (Fig. 3B, C). Further, Ptgs2 expression is significantly lower in the M-pole of p53 deficient (Trp53$^{f/f}$ Pgr$^{Cre/+}$, Trp53$^{d/d}$) mice, suggesting that Trp53$^{d/d}$ mice have an overall lower level of COX2 compared to Trp53$^{f/f}$ mice (Fig. 3C).

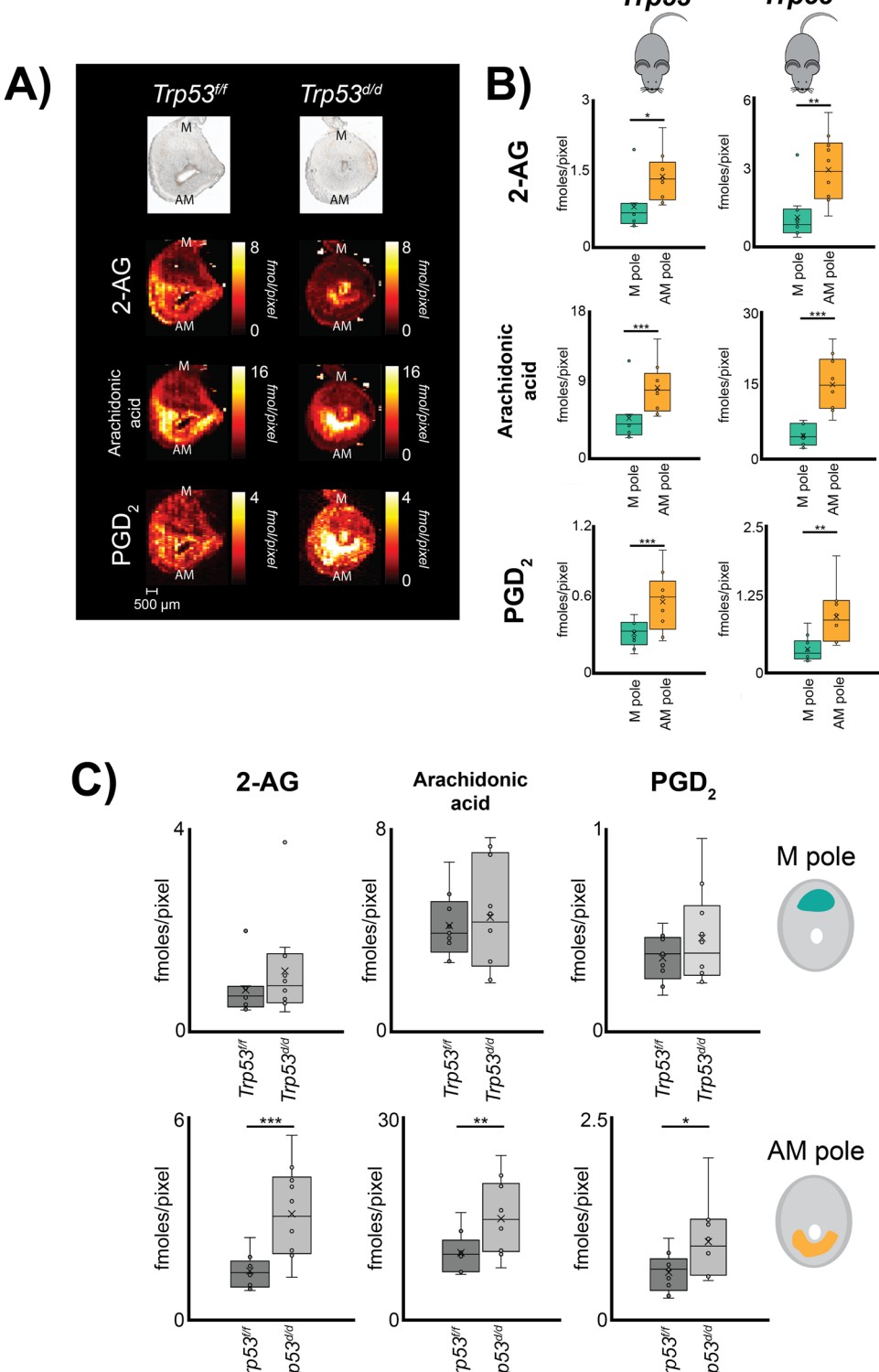

**Fig. 2 Distribution of prostaglandins to morphological regions in mouse embryo implantation sites on day 8 of pregnancy using quantitative silver doped nano-DESI MSI. A** Optical images and corresponding quantitative ion images of 2-AG, arachidonic acid, and PGD2 in *Trp53f/f* and *Trp53d/d* mice on day 8 of pregnancy. Color scaling corresponds to detected fmoles/pixel with the values going from dark to bright. Note that the fixed quantitative color scales for each metabolite facilitates direct visual comparisons. Scale bar 500 μm. M mesometrial pole; AM anti-mesometrial pole. **B** Regions of interest for the AM-pole and M-pole of *Trp53f/f* and *Trp53d/d* mice, respectively, with detected concentrations of 2-AG, arachidonic acid, and PGD2 from three biological and three technical replicates. **C** Region of interest analysis of the AM-pole and M-pole for 2-AG, AA, and PGD2 of 3 biological and 3 technical replicates comparing the of *Trp53f/f* and *Trp53d/d* mice (*n* = 9 for each genotype). Error bars in represent 1 standard deviation and * *p*-value < 0.05, ** *p*-value < 0.01, *** *p*-value < 0.001.

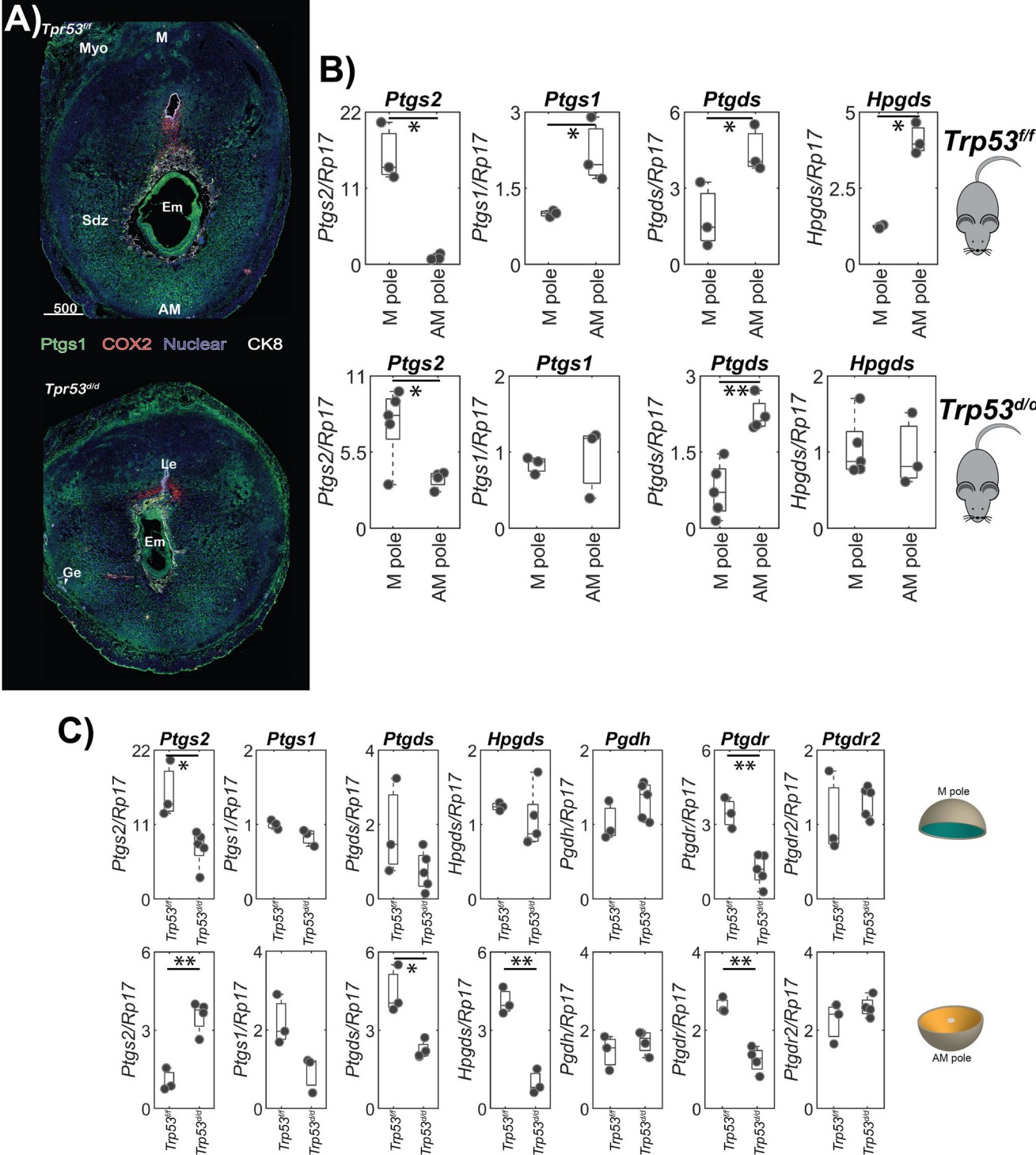

**Fig. 3 Distribution of proteins involved in PG synthesis, hydrolysis, and signaling in implantation sites on day 8 of pregnancy. A** Fluorescence in situ hybridization of *Ptgs1* and COX2 in *Trp53*[f/f] and *Trp53*[d/d] implantation sites on day 8 of pregnancy. The embryonic trophoblasts are outlined by Cytokeratin 8 (CK8) immunostaining. M mesometrial pole; AM anti-mesometrial pole; Em embryo; SDZ secondary decidual zone; Myo myometrium; scale bar, 500 μm. **B** Segmented RT-qPCR results for *Ptgs2*, *Ptgs1*, *Ptgds*, *Hpgds*, in the AM and M-poles in *Trp53*[f/f] and *Trp53*[d/d] implantation sites on day 8 of pregnancy. **C** Segmented RT-qPCR results showing the expression of *Ptgs2*, *Ptgs1*, *Ptgds* and *Hpgds, Pgdh, Ptgdr, and Ptgdr2* in the M-pole and AM-pole of the implantation sites comparing the *Trp53*[f/f] and *Trp53*[d/d] implantation sites. Significance is determined by a two-tailed student's *t*-test, where * *p*-value < 0.05, ** *p*-value < 0.01, *** *p*-value < 0.001.

These results illustrate a clear, region-specific perturbation of COX2 expression levels in $Trp53^{d/d}$ mice. FISH imaging corroborates the results from RT-qPCR, showing that $Ptgs1$ is expressed on decidual cells, mainly in the secondary decidual zone on the AM-pole, for both $Trp53^{d/d}$ and $Trp53^{f/f}$ mice, while $Ptgs2$ is mainly expressed in the M-pole (Fig. 3A).

Our RT-qPCR data reveal an opposing impact of p53 on PGD$_2$ synthases when compared to $Ptgs2$ (Fig. 3C). The expression of $Ptgds$ and $Hpgds$ are significantly decreased in the AM-pole of the $Trp53^{d/d}$ mice as compared to control mice; however, the relative levels of $Ptgds$ in the AM-pole are still ~2-fold higher than in the M-pole (Fig. 3C). In the M-pole, the $Ptgds$ and $Hpgds$ levels of $Trp53^{d/d}$ mice are similar to the $Trp53^{f/f}$ control mice.

The RT-qPCR data indicates a crucial role of p53 in controlling PG synthesis by regulating both $Ptgds$ and $Hpgds$ levels in the AM-pole (Fig. 3C). Segmented RT-qPCR analysis of $Ptgdr$ and $Ptgdr2$ expression levels, indicating the abundance of PGD$_2$ receptors, show that $Ptgdr$ is ~3 times lower in both the AM and M-pole of the $Trp53^{d/d}$ mice, compared to that of $Trp53^{f/f}$ mice (Fig. 3C). In addition, no significant change in Pgdh expression was observed between $Trp53^{f/f}$ and $Trp53^{d/d}$ mice suggesting no alterations in PG dehydrogenase levels resulting from conditional uterine deletion of p53. (Fig. 3C).

## Discussion

Technical limitations in direct detection of PGs in tissue sections largely arise from the inability to selectively stain individual PGs for in situ localization studies with fluorescent microscopy. Further, the low endogenous abundance and poor MS ionization efficiency of PGs have hindered the use of MSI techniques for mapping their distribution in tissue. The inclusion of silver ions in the nano-DESI solvent increases the ionization efficiency of PGs and enables direct visualization of PGs in tissue. However, the identification of specific PGs in selected domains remains uncharted territory. We combined our novel PG MSI method with traditional RNA in situ hybridization, making it possible to identify specific PGs in situ, which significantly advances our current understanding of PG signaling in uterine biology and beyond.

The main localization of PG and PG precursors is to the AM-pole. Thus, at day 8 of pregnancy, it is likely that $Ptgds$ and PGD$_2$ localization to the AM-pole signifies a defined role in AM-pole decidualization. Our previous study also suggests PGD$_2$ and PGE$_2$ are major players in endometrial cancer[33]. Although the role of PGD$_2$ remains unclear in the mouse uterus, cells with positive PGD$_2$ signals undergo tissue remodeling to make room for growing fetuses. There are several known downstream effects of PGD$_2$ signaling, which include inhibition of platelet aggregation and relaxation of vascular and non-vascular smooth muscle[34,35]. Therefore, PGD$_2$ localization to the lateral and AM-pole on day 8 suggests its importance to ensure adequate nutrient delivery to the developing embryo and facilitate tissue reorganization.

Localization of PG precursors and PGD$_2$ to the AM-pole of the implantation site nicely correlates with the expression of the PGD$_2$ synthases $Ptgds$ and $Hpgds$ in implantation sites on day 8 of pregnancy. This localization further correlates with the expression of $Ptgs1$ encoding for the housekeeping enzyme COX1 in agreement with our previous results[36]. However, PGD$_2$ localization is in contrast with the expression and localization of COX2, which is primarily observed in the M-pole, suggesting that COX1 plays an important role in decidualization and parturition. Although our current results show overlapping localization of PGD$_2$, arachidonic acid, and 2-AG with $Ptgs1$, the role of COX1 is obscured by the fact that $Ptgs2$ can compensate for $Ptgs1$ loss[11,13]. Specifically, although $Ptgs1^{-/-}$ mice show no obvious defects in early

pregnancy[13], our data of $Ptgs1$ and PGD$_2$ suggests that COX1 plays a role in early pregnancy. Further, COX2 is the major inducible enzyme for the first rate-limiting step in the biosynthesis of PGs, and disruption of its function causes multiple female reproductive failures, including infertility in $Ptgs2^{-/-}$ mice[11]. Therefore, we hypothesize that $Ptgs2$ is induced to compensate for COX1 in $Ptgs1^{-/-}$ mice, so that no apparent defects are observed in the latter. In fact, our previous study supports this hypothesis, in that $Ptgs2$ is expressed in luminal epithelia in $Ptgs1^{-/-}$ mice on day 4 of pregnancy where $Ptgs1$ is normally expressed in wild-type mice[14].

The role of COX appears pleiotropic during pregnancy[9]. $Ptgs2$ is subject to cell-specific expression in the mouse uterus depending on the experimental conditions and implantation-specific gene expression[37–39]. The excess of available arachidonic acid in the AM-pole corroborates that COX is indeed the rate-limiting step in PG synthesis, in agreement with the previous studies[3,4]. Further, the lower relative abundance of 2-AG to arachidonic acid suggests that arachidonic acid does not exclusively originate from 2-AG, but rather from a larger diverse pool of lipids, such as phospholipids and diacylglycerols. This is further evidenced by our previous study from day 8 implantation sites reporting a clear localization of arachidonic acid-containing phospholipids and diacylglycerols to the AM-pole[20]. In addition to PGD$_2$, PGF$_{2\alpha}$ localizes to the myometrium of the uterus matching the expression of $Ptgs1$ encoding COX1, again away from COX2 expression and localization. Although $Ptgs1$ is considered to be a housekeeping gene that maintains basal cell functions, the disparate expression of COX2 with PG and PG precursor localization indicates that COX2 abundance does not directly correlate with in situ PG distributions in tissue.

Conditional deletion of $Trp53$ is known to impact pregnancy progression of mice[18]. Although the embryo development is compromised in $Trp53^{d/d}$ mice, the overall expression pattern and intensity of $Ptgs1$ are comparable to those in $Trp53^{f/f}$ mice. An increase of COX2 in the AM-pole with a concurrent decrease in the M-pole suggests that the absence of p53 causes a significantly arrested transition of $Ptgs2$ from the AM-pole to the M-pole between days 5 and 8 of pregnancy[36].

We observed an increase of PGD$_2$ along with 2-AG and arachidonic acid in decidual cells at the anti-mesometrial side is observed in WT mice on day 8 of pregnancy. The physiological roles of PGD$_2$ in decidualization are still unclear. Our earlier study revealed that oxidized lipids were found to be decreased in the AM-pole of $Trp53^{d/d}$ implantation sites, indicating lower levels of reactive oxygen species in p53 deficient uteri[20]. As a result, the higher PGD$_2$ levels in $Trp53^{d/d}$ mice are not likely to originate from autoxidation of arachidonic acid through the isoprostane pathway, but rather from elevated levels of COX2 in the AM-pole. Accumulation of PGD$_2$ and arachidonic acid in the AM-pole of $Trp53^{d/d}$ mice is accompanied with increased polyploid cells and decidual cell senescence[18], indicating PGD$_2$'s role in decidual aging and/or differentiation. It would be interesting to examine whether the elevated PG levels in $Trp53^{d/d}$ decidua extend to parturition, since PGs are closely associated with parturition timing[17] and contribute to spontaneous preterm birth in $Trp53^{d/d}$ females[18].

The higher abundance of PGD$_2$ observed in the AM-pole of the $Trp53^{d/d}$ mice might stimulate higher $Pgdh$ activity. However, since no significant change in $Pgdh$ expression was observed between the M-poles and AM-poles of $Trp53^{f/f}$ and $Trp53^{d/d}$ mice, the increased levels of PGD$_2$ could signify an increase in PGD$_2$ signaling in the AM-pole of the $Trp53^{d/d}$ mice. PGD$_2$ has two primary receptors, DP1 and DP2, which are known to have antagonistic roles in regulating cAMP levels and calcium mobilization[40,41]. In combination with the increased PGD$_2$ levels

in the AM-pole of the $Trp53^{d/d}$ mice, the lower levels of DP1 in the AM-pole could impact the balance of cAMP and calcium mobilization through $PGD_2$ signaling.

Spatial information of specific PGs greatly advances our understanding of COX/PG signaling. For example, whole tissue digests would have overlooked elevated $PGD_2$ levels in specific cellular regions such as the AM-pole, and the factual increase in this region would have been diluted throughout the rest of the tissue. Further, RT-qPCR of homogenized whole uterine implantation sites would have yielded an overall lower level of COX2 in the $Trp53^{d/d}$ mice. As a result, the higher abundance of COX2 in the AM-pole would have been overlooked. The lower observed abundance of COX2 in combination with lower levels of $Ptgds$ and $Hpgds$ in $Trp53^{d/d}$ mice would have suggested that $PGD_2$ was decreased in the $Trp53^{d/d}$ mice by RT-qPCR alone. Yet the combination of small molecule and synthetic enzyme spatial information reveals that $PGD_2$ is indeed higher in abundance in the $Trp53^{d/d}$ mice, despite the lower $Ptgds$ and $Hpgds$ levels in this region. These results highlight the importance of both COX1 and COX2 in PG biosynthesis.

In this study, MSI was employed to detect PGs, but only $PGD_2$ and $PGF_{2\alpha}$ were detected. This means that these are the most abundant PG species in implantation sites at day 8 of pregnancy, and suggests that they play an important role in the early stages of pregnancy. An earlier study observed $PGI_2$ synthase localizing to the M-pole of day 8 mouse uterus implantation sites, indicating a role also for $PGI_2$ at that stage in early placentation[12]. We hypothesize that the biosynthesis of PG species that are below the limit of detection in this study would be impacted by the nearly two times lower expression of COX2 in the M-pole of the $Trp53^{d/d}$ mice, thereby contributing to the compromised embryonic development in this genotype. These results in combination with the importance of PGs in cell physiology and signaling will drive us to develop additional measures for increasing PG coverage for in situ localization in tissue.

## Conclusion

The most abundant prostaglandin species detected in mouse uterine implantation sites on day 8 of pregnancy was $PGD_2$, which co-localized with PG precursor metabolites and $PGD_2$ synthases in the AM-pole. Conversely, $Ptgs2$ levels were found to be 15 times higher in the M-pole, which restricts its use as a marker for prostaglandin localization. Our data revealed that COX1 actually co-localized with $PGD_2$ and $PGF_{2\alpha}$ on day 8 of pregnancy. Further, upon conditional deletion of p53 in mouse uterine tissue, PG synthesis was elevated in the AM-pole, which was concomitant with higher expression of $Ptgs2$. However, despite the higher levels of $PGD_2$ and $Ptgs2$ in the AM-pole of p53 deficient mice, the opposite was true $Ptgds$ and $Hgpds2$, which showed reduced mRNA expression. Thus, no single synthase or combination of synthase enzymes was fully indicative of in situ PG abundance in tissue. Despite only monitoring $PGD_2$ and PG synthases in detail, we anticipate these data can be interpolated to suggest that similar discrepancies may exist for PG species not detected in this study due to mRNA for the rate-limiting enzymes $Ptgs1$ and $Ptgs2$ not mirroring PG abundance. Altogether, our results indicate the importance of COX1 in PG biosynthesis, and the importance of in situ detection of each PG species to fully elucidate PG biosynthesis.

## Materials and methods

**Mice.** $Trp53^{loxP/loxP}Pgr^{Cre/+}$ ($Trp53^{d/d}$) mice were generated as described[18]. All genetically modified mice and WT controls were housed in the animal care facility at the Cincinnati Children's Hospital Medical Center according to NIH and institutional guidelines for laboratory animals. All protocols of the present study were approved by the Cincinnati Children's Hospital Research Foundation Institutional Animal Care and Use Committee. All mice were housed in wall-mount negative airflow polycarbonate cages with corn cob bedding. They were provided ad libitum with double distilled autoclaved water and a rodent diet (LabDiet 5010). Female mice were mated with WT fertile males to induce pregnancy (vaginal plug = day 1 of pregnancy). Mice were euthanized by cervical dislocation right before tissue collection under deep anesthesia. Mice were killed on day 8 of pregnancy.

**Fluorescence in situ hybridization (FISH).** In situ hybridization was performed as previously described[42]. In brief, implantation sites from three individual animals in each experimental group were collected. Frozen sections (12 µm) from three implantation sites from different females in each group were mounted onto poly-L-lysine-coated slides and fixed in 4% paraformaldehyde in PBS. Following acetylation and permeabilization, slides were hybridized with the DIG-labeled $Ptgs1$ at 55 °C overnight. After hybridization, slides were then washed, quenched in $H_2O_2$ (3%), and blocked in blocking buffer (1%). Anti-Dig-peroxidase was applied onto hybridized slides and color was developed by Tyramide signal amplification (TSA) Fluorescein according to the manufacturer's instructions (PerkinElmer). The slides were then immunostained with antibody for Cytokeratin 8 (Iowa hybridoma bank, 1:100 dilution). Alexa 594 conjugated secondary antibodies (used in 1:400 dilution) were from Jackson Immunoresearch. Nuclear staining was performed using Hoechst 33342 (H1399, Molecular Probes, 2 µg/ml). Immunofluorescence was visualized under a confocal microscope (Nikon Eclipse TE2000).

**Segmented quantitative RT-PCR.** Mesometrial and antimesometrial tissues were separated by the top edge of implantation chambers. The whole implantation site was separated into two halves along the M−AM axis, and the implantation chamber with an embryo was exposed. M and AM poles were obtained by cutting a half of the implantation site following a cutting plane vertical to the M−AM axis at the top edge (close to the M pole) of the implantation chamber. The embryonic tissues were removed from the AM pole prior to RNA was collected from $Trp53^{f/f}$ and $Trp53^{d/d}$ implantation sites. RNA was analyzed as previously described[43]. In brief, total RNA was extracted with Trizol (Invitrogen, USA) according to the manufacturer's protocol. After DNase treatment (Ambion, USA), 1 µg of total RNA was reverse transcribed with Superscript II (Invitrogen). Quantitative PCR was performed using StepOne™ Real-Time PCR System. All data presented were normalized against levels of $rPl7$ which served as an internal loading control. For statistical analysis between samples, a two-tailed F-test was carried out to determine equal or unequal sample variances. Following the corresponding two-tailed student's t-test was used to determine statistical differences between samples.

**Nano-DESI MSI of uterine implantation sites.** Thin tissue sections from day 8 mouse uterine implantation sites for three $Trp53^{f/f}$ and three $Trp53^{d/d}$ mice were mounted on regular glass slides and subjected to quantitative sliver doped nano-DESI MSI[28]. At least three technical replicates from each mouse were conducted, totaling at least nine replicate images for both $Trp53^{f/f}$ and $Trp53^{d/d}$ mice. Nano-DESI images were acquired using 150 µm OD and 50 µm ID primary and secondary capillaries with an experimental set-up described in an earlier manuscript[29]. The nano-DESI solvent consisted of 9:1 acetonitrile:methanol (LC-MS grade, Merck, Germany), doped with 10 ppm $Ag^+$(AgNO3, > 99%, Fisher Scientific, Gothenburg, Sweden), 0.1% formic acid (100%, Merck), 2.5 µM arachidonic acid-d8 (99% D, Merck), and 0.5 µM $PGF_{2\alpha}$-d9 (99% D, Merck), and was propelled through the nano-DESI capillary interface at 0.5 µL/min. To acquire mass spectrometry images, motorized stages propelled the sample under the nano-DESI capillary interface at 20 µm/s along the x-axis, and stepped along the y-axis in 150 µm increments. Mass spectrometry data were acquired using a QExactive MS (Thermofisher Scientific, Bremen, Germany) in positive ion mode, with an electrospray voltage of 3 kV, a heated capillary temperature of 250 °C, an AGC target of $1 \times 10^6$, and a maximum ion accumulation time of 200 ms. Two scan functions were stacked back to back—one narrow window full scan targeted for prostaglandin silver adduct detection between m/z 458–473, and a broader range full scan for untargeted investigations between m/z 300 and 700. Overall, the MS duty cycle for each scan and sample motion under the nano-DESI capillary interface generated pixels roughly 150 × 20 µm.

**MSI data processing.** All data processing was carried out using custom scripts, where Thermofisher RAW files were first converted to centroided mzXML files for direct import into Matlab. To generate ion images, the closest m/z peak of interest obtained from a targeted list was selected within a 5 ppm mass window, and m/z peaks were aligned in time along the x-axis to account for different numbers of scans for each line scan on the y-axis (Table S4). For quantitative comparisons of MSI data, endogenous $PGD_2$ and $PGF_{2\alpha}$ were determined using a 0.5 µM $PGF_{2\alpha}$-d9 internal standard added to the nano-DESI solvent. PG precursors 2-AG and FA 20:0 were quantified from a 2.5 µM FA 20:4-d8 internal standard. Regions of interest were extracted with custom scripts and AM-pole, M-pole, and myometrium anatomical regions were drawn based on the ion image for arachidonic acid. Internal standard normalized average intensities were extracted for each ROI of the ion image, and the top and bottom 1 percentile of pixels were removed to minimize extreme values. Arachidonic acid and 2-AG were normalized to arachidonic acid-

$d_8$, and $PGD_2$ and $PGF_{2\alpha}$ were normalized to $PGF_{2\alpha}$-$d_9$. To compare values across replicate mass spectrometry images technical replicates were combined with biological replicates, and an equal variance two-tailed student's t-test was employed.

**LC-IMS-MS analyses**. Lipids were extracted from the uteri tissue using a Folch extraction[44], where the tissue was lysed in 400 μl methanol using a tissue lyser with a 3 mm tungsten carbide bead for 3 min. Samples were then transferred into a vial and 800 μl of chloroform was added. The sample was vortexed for 60 s then shaken for 1 h at room temperature at 1000 rpm. The samples were then vortexed again briefly and 300 μl of water was added to induce a bi-phase separation. The sample was gently mixed, incubated at room temperature for 10 min, and then centrifuged for 5 min at $15,000 \times g$ at 4 °C. The lower organic layer was collected and transferred into another vial. The remaining sample was washed using a blank lower organic layer to collect the remaining lipids. The washed samples were vortexed for 5 s, incubated at room temperature for 10 min, and then centrifuged as stated above. The lower organic layer of the washed sample was added to the first lower organic layer and then dried *in vacuo*. One hundred fifty microliters of 2:1 chloroform/methanol was added and the samples were stored at −20 °C until analysis.

The LC analyses of the uteri tissue were performed using a Waters NanoAquity UPLC system interfaced with the 6560 IMS-QTOF MS instrument (Agilent, Santa Clara, CA)[45]. Initially, the lipid extracts were dried *in vacuo*, reconstituted in 200 μL of isopropanol, and analyzed with LC-IMS-MS. Here 0.7 μL of the sample was injected onto a capillary column (26 cm × 150 μm i.d.) containing HSS T3 reversed-phase material (1.8 μm). Lipids were separated over 90-min using gradient elution (mobile phase A: acetonitrile/water (40:60) containing 10 mM ammonium acetate; mobile phase B: acetonitrile/isopropanol (10:90) containing 10 mM ammonium acetate) at a flow rate of 1 μl/min. The IMS-MS analyses were then performed in both positive and negative ionization and collected from 100 to 3200 m/z at a resolution of 40,000.

**Statistics and reproducibility**. For RT-qPCR analysis, implantation sites were dissected and segmented from 3 $Trp53^{f/f}$ and 5 $Trp53^{d/d}$ mice. In total, mRNA was extracted and quantified from the AM and M poles of one implantation site from each animal. An F-test was employed to determine the equal or unequal variance between $Trp53^{f/f}$ and $Trp53^{d/d}$ data, followed by the corresponding two-tailed student's t-test.

For MSI experiments, implantation sites from three $Trp53^{f/f}$ and three $Trp53^{d/d}$ mice were extracted to capture biological variance. At least three technical replicates were performed for each mouse, totaling nine analyzed sections for the $Trp53^{f/f}$ implantation sites, and ten analyzed sections for $Trp53^{d/d}$ implantation sites. Regions of interest analysis included data from all analyzed sections, where an F-test was used to confirm equal variance between sample sets, followed by a two-tailed student's t-test.

**Reporting summary**. Further information on research design is available in the Nature Research Reporting Summary linked to this article.

## Data availability

All data needed to evaluate the conclusions in the paper are present in the paper and/or the Supplementary Materials. Source data used to generate the figures are available in the Supplementary Data 1 Excel file. Any remaining information can be obtained from the corresponding author upon reasonable request.

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

## Acknowledgements

This research project was provided by the Swedish Foundation for Strategic Research (IL), the Swedish Research Council (IL), National Institute on Drug Abuse (DA006668) (SKD), and Eunice Kennedy Shriver National Institute of Child Health and Human Development (HD068524 & HD103475) (SKD). The authors acknowledge PNNL ion mobility analyses were supported by the National Institutes of Health (NIH) Eunice Kennedy Shriver National Institute of Child Health and Human Development grant R21 HD084788.

## Author contributions

K.D., X.S., S.D., and I.L. designed the study; K.D. collected the MSI data; X.S. collected the biological material, performed the PCR and FISH experiments; E.S.B. collected the LC-IMS-MS data; K.D. and X.S performed the data analysis, all authors discussed the results and prepared the paper.

## Funding

## Competing interests

The authors declare no competing interests.
