## [Peer Review File · Communications Biology]

Reviewers' comments:

Reviewer #1 (Remarks to the Author):

The manuscript entitled "In situ imaging reveals disparity between prostaglandin localization and abundance of prostaglandin synthases" by K. D. Duncan and colleagues, describes the localization of prostaglandins (PG) to mouse embryo implantation sites on day 8 of pregnancy, in parallel to PG synthase expression.

Prostaglandins (PGs) are important lipid mediators in numerous physiological processes. They are present in tissues at very low concentrations; so far, they have been detected by immunoassays or liquid chromatography-mass spectrometry (LC-MS). Recently, the use of mass spectrometry imaging (MSI) allowed the direct MS analysis of the distribution and abundance of various small molecules or lipid species in tissue.

The challenge of this study is to image PG species to specific cellular regions, using an improved technique of MSI, the silver-doped nanospray desorption electrospray ionization MSI (nano-DESI). This technique has been previously set up by the same authors to enable PG localization in mouse uterine tissue sections (Duncan et al, 2018, Anal. Chemistry).

In this study, the nano-DESI imaging method allows the direct in situ tissue localization of prostaglandins and combination with RNA in situ hybridization and RT-qPCR techniques, shows that prostaglandin abundance and spatial distribution cannot be determined by monitoring only PG synthases. Finally, as uterine-specific deletion of p53 leads to aberrations in COX2-derived prostaglandin signaling, this approach is applied to study the impact of p53 on PG biosynthesis in early embryonic development.

This study is highly novel and interesting; it calls into question the numerous approaches identifying the expression profiles of synthases that conclude on the localization of the production of prostaglandins. Until now, localization of PG species in specific tissues was deduced from the spatiotemporal pattern of PG synthases and the measurable accumulation of PGs in these tissues and were quantified by ELISA or LC-MS.

However, there are several concerns that should be revised.

1- Page 3 lanes 34-35: this sentence is not completely true and should be modified; PG-biotin has been used and revealed by the avidin-biotin peroxidase complex system, to visualize PGE2 or PGF2 α in tissue (Siegel et al. Lab. Investigation 1987; Miyauchi et al. Histochem. Cell Biology 1996).

2- Page 5 lanes 4, 9 and 17: Please correct description of Figure 1 in the text and modify this figure; remove Figure 1B, Figure 1D does not describe qPCR experiments, Figure 1E in the text is not shown on Figure 1.

3- Page 6 and Figure 2A: modify panel Ptges/Ptgds ; Ptgds is written in the text (lane 16).
Figure 2D: show quantification of PGF2 α .

4- Figure 2: why authors focused on PGD2 and PGD2 synthases while PGE2 was reported as a major PG in the uterine physiopathology. Particularly, Cox-2 expression is in a positive feedback loop with its products PGE2 and PGF2 α ; PGE2, PGF2 α , PGI2 and its metabolite 6-keto-PGF α 1 have been significantly measured in uterine tissue.

PGF synthase mRNA expression should be shown and quantitative nano-DESI of PGE2 should be performed. Together with Figure 2, Figure S1 should be mentioned in this section.

5- Figure 3: data should be supplemented with PGF synthase mRNA expression and nano-DESI of PGE2 and PGF2 α . Previous study showed that deletion of uterine Trp53 induced preterm birth through a COX2/PGF synthase/PGF2 α pathway (Hirota et al. 2010).

These points 4- and 5- are main concerns as title of the manuscript, subtitles of the Results and Discussion sections contain the general terms of PG localization, PG synthases, PG synthesis, PG biosynthesis....

6- Panels showing COX2 FISH (Figures 2B and 3B) should be enlarged.

7- There are many inaccuracies in the numbering and the legends of figures.

Page 8-9: the text refers to Figure 3 instead of Figure 2.

Figure S1 has an incomplete legend.

8- Figure S1: these results are only partially described in the Result section of the manuscript (Fig S1 is referred on page 5 but not on page 7).

The X axis should be modified according to that of Figure 2A panels (AM/M poles).

Why authors used another combination of synthases (Ptges, Ptges3 and not H-Pgds) and analyzed PGD2 receptor expression?

Some of the panels are redundant with panels of other figures (ex: Ptgds on Fig S1 and Fig. 2A/ Fig. 3A).

9- The term qPCR is used all along the manuscript instead of RT-qPCR.

10- Authors should discuss why only PGD2 and PGF2a were detected using nano-DESI while Ptges is highly expressed at the AM pole, at a similar level to that of both Ptgds.

11- Another main concern in this manuscript, is the lack of information on statistical analyses (authors should refer to the authors guidelines of the Communications Biology journal).

“The Methods must include a statistics and reproducibility section with the following information: the name of the statistical test, the n value for each statistical analysis, the comparisons of interest, a justification for the use of that test... Authors must state whether a number that follows the \pm sign is a standard error (s.e.m.) or a standard deviation (s.d.).

Legends of Figures 2 and 3 do contain some informations; however, the statistical unit is not defined. Do

three biological replicates mean three embryos from the same litter or from 3 different litters?

Reviewer #2 (Remarks to the Author):

Duncan et al. used a novel mass spectrometry imaging method to assess the localization and concentration of prostaglandins (PGs) in mouse tissues during pregnancy. Although the study is descriptive, this new method, a quantitative nano-DESI MSI, has clear advantages over whole tissue MS/MS analysis. This study should be of interest to Communications Biology's readers. It brings new biological insights on the spatial evaluation of PGs in tissues, which could reveal the functional significance of each PGs during different stages of embryo development/pregnancy. Although this work allows advancement in the field, there are several concerns that have to be addressed, as listed below.

Major concerns:

1. As this is a new method, the authors need to include a negative control (to control for a false-positive result). To strengthen the findings, the authors need to include a negative control in Figure 2, i.e., uterine tissues from non-pregnant mice.
2. The authors need to provide the clear goal of the study, to visualize PGD2 or to provide a new method? Or both?
3. The authors showed that *Ptgs2* mRNA levels were significantly higher in Trp53d/d in the AM, whereas the levels were lower in the M compared to Trp53f/f (Fig. 3A). However, it is difficult to see this result in the expression in the tissues in Figure 3B. To this reviewer, levels of COX2 in Figure 3B are similar between Trp53d/d and Trp53f/f tissues. Please provide larger images or zoomed-in images as insets.
4. Please use correct nomenclature throughout the manuscript, including the text and the figures, i.e., *Ptgs1* and *Ptgs2*, italicized for mRNA and all capitalized, non-italicized for proteins. It is confusing at times when trying to interpret the results in Figures 2 and 3.
5. The explanation for the significance of anti-mesometrium vs. mesometrium needs to be introduced in the introduction, instead of the discussion (page 8, lines 14-19). This is important as the readers might not be familiar with the specifics of these two regions. In addition, not much of an explanation was included for the results in Figure 1.
6. Statement on page 8, lines 38-43, if that is the case, the authors need to include the measurement of PGD2 level in *Ptgs2*^{-/-} uterus. The PDG2 levels should be comparable between *Ptgs2*^{-/-} and *Ptgs2*^{+/+} uteri.
7. Include the justification for the assessment of tissues from Trp53d/d mice on day 8 of pregnancy. This could confuse the readers as the authors mentioned on page 4, line 11 that 'deletion of p53 is elevated levels of COX2 activity on day 16 of pregnancy'.
8. Please include the type of statistical analyses for all tests in the manuscript.
9. Please briefly explain how AM & M tissues were isolated. Were there any transcripts (markers) were measured for the purity of the isolation?
10. Please label "M" and "AM" on images in Figures 2C and 3C
11. Please plot the raw value for each individual data point (in addition to mean) for Figures 2D and 3D.

12. Please consider presenting the data from M before AM as the tissues orient from top to bottom. This comment applies to Figures 3A and 3D. (switching data from M to the top and AM to the bottom panel)

Minor comments:

1. Some of “2” in PGD2 were subscripts, sometimes there were not. Please be consistent throughout the manuscript.

Reviewer #3 (Remarks to the Author):

The authors used silver-doped nanospray desorption electrospray ionization (nano-DESI) MSI to reveal in situ localization and abundance of prostaglandin species along with situ localization of prostaglandin synthetic enzymes in mouse embryo implantation sites on day 8 of pregnancy. They found PGD2, its precursors, and downstream synthases co-localize with the highest expression of COX1, but not COX2. Further, they found that the abundance of COX and prostaglandin D2 synthases in cellular regions does not mirror the regional abundance of prostaglandins when conditional deletion of transformation-related protein 53, a known regulator of COX2-derived prostaglandin signaling, was carried out. They deduce that prostaglandins tissue localization and abundance may not be inferred by COX or prostaglandin synthases in uterine tissue, and must be resolved by an in situ prostaglandin imaging. Their findings, particularly the part on PGD2, are interesting and may be of interest to reproductive biologists to study role of PGD2 in implantation and early development. However, there are some concerns which requires the authors' attention.

Major:

1. It may be true that mere measurements of PG synthetic enzymes may not mirror the abundance of a particular PG specie. However, we should also bear in mind that PGs are quite diffusible and their abundance at a particular site is determined not only by their synthesis but also by their diffusion and degradation. I suggest the authors consider the diffusion factor and look at the enzyme responsible for their degradation at the same time.

2. It is well known that a number of other PGs are involved in implantation as well. The authors focused mainly on PGD2 and PGF2alpha in this study. I wonder whether the claimed phenomena also hold true in other PG species.

3. Mostly, the authors used mRNA abundance to reflect the abundance of PG synthetic enzymes. Apparently, mRNA level may not necessarily reflect protein level. Is it possible to measure the protein level at the same time?

Minor:

1. For the abstract, the authors should introduce the background on p53 in terms of PG synthesis. Otherwise, it comes out of blue and is hard to understand.

2. Because the methodology is also an important part of the manuscript, I would suggest that the authors put all supplementary figures into the manuscript.

3. Can you show the data that only PGD2 and PGF2α were detected by MSI ?

Reviewer #1 (Remarks to the Author):

The manuscript entitled “In situ imaging reveals disparity between prostaglandin localization and abundance of prostaglandin synthases” by K. D. Duncan and colleagues, describes the localization of prostaglandins (PG) to mouse embryo implantation sites on day 8 of pregnancy, in parallel to PG synthase expression.

Prostaglandins (PGs) are important lipid mediators in numerous physiological processes. They are present in tissues at very low concentrations; so far, they have been detected by immunoassays or liquid chromatography-mass spectrometry (LC-MS). Recently, the use of mass spectrometry imaging (MSI) allowed the direct MS analysis of the distribution and abundance of various small molecules or lipid species in tissue.

The challenge of this study is to image PG species to specific cellular regions, using an improved technique of MSI, the silver-doped nanospray desorption electrospray ionization MSI (nano-DESI). This technique has been previously set up by the same authors to enable PG localization in mouse uterine tissue sections (Duncan et al, 2018, Anal. Chemistry).

In this study, the nano-DESI imaging method allows the direct in situ tissue localization of prostaglandins and combination with RNA in situ hybridization and RT-qPCR techniques, shows that prostaglandin abundance and spatial distribution cannot be determined by monitoring only PG synthases. Finally, as uterine-specific deletion of p53 leads to aberrations in COX2-derived prostaglandin signaling, this approach is applied to study the impact of p53 on PG biosynthesis in early embryonic development.

This study is highly novel and interesting; it calls into question the numerous approaches identifying the expression profiles of synthases that conclude on the localization of the production of prostaglandins. Until now, localization of PG species in specific tissues was deduced from the spatiotemporal pattern of PG synthases and the measurable accumulation of PGs in these tissues and were quantified by ELISA or LC-MS.

However, there are several concerns that should be revised.

1- Page 3 lines 34-35: this sentence is not completely true and should be modified; PG-biotin has been used and revealed by the avidin-biotin peroxidase complex system, to visualize PGE2 or PGF2 α in tissue (Siegel et al. Lab. Investigation 1987; Miyauchi et al. Histochem. Cell Biology 1996).

Reply: We thank the reviewer for this clarification and have amended the sentence as follows:

“Despite the biological importance of PGs, localization of native PGs to specific cells in tissues has not yet been possible, although methods using PGs labelled with biotins have helped in this regard (Siegel et al and Miyauchi et al).”

2- Page 5 lines 4, 9 and 17: Please correct description of Figure 1 in the text and modify this figure; remove Figure 1B, Figure 1D does not describe qPCR experiments, Figure 1E in the text is not shown on Figure 1.

Reply: We thank the reviewer for recognizing these mistakes and have amended the text accordingly.

3- Page 6 and Figure 2A: modify panel Ptges/Ptgds ; Ptgds is written in the text (line 16).
Figure 2D: show quantification of PGF2 α .

Reply: We have modified the figures 2 and 3 to provide a better flow for the reader and added the ion images of PGF2a to the supporting information. The number of pixels with detected signal was not sufficient for regions of interest analysis in the AM pole and M pole of for PGF2a. However, we have added a graph in the supporting information showing the quantification of PGF2alpha in the myometrium for Trp53f/f and Trp53d/d mice.

4- Figure 2: why authors focused on PGD2 and PGD2 synthases while PGE2 was reported as a major PG in the uterine physiopathology. Particularly, Cox-2 expression is in a positive feedback loop with its products PGE2 and PGF2 α ; PGE2, PGF2 α , PGI2 and its metabolite 6-keto-PGF α 1 have been significantly measured in uterine tissue. PGF synthase mRNA expression should be shown and quantitative nano-DESI of PGE2 should be performed. Together with Figure 2, Figure S1 should be mentioned in this section.

Reply: We agree with the reviewer that it would be nice to also have data on PGE2, however, we only detected PGD2, which suggests that PGE2 levels are below our limit of detection. Despite our intension, we did not detect any of the other PG species either and therefore we focus on the detected PGD2 species and its synthases. The text has been updated to also mention Figure S1.

5- Figure 3: data should be supplemented with PGF synthase mRNA expression and nano-DESI of PGE2 and PGF2 α . Previous study showed that deletion of uterine Trp53 induced preterm birth through a COX2/PGF synthase/PGF2 α pathway (Hirota et al. 2010).

Reply: As stated above, PGE2 was below the limit of detection in the AM and M poles, and because of the low abundance of PGF2a we suspect that we are not capturing its full dynamics. In addition, the main abundance of PGF2a in the myometrium instead of the M-pole and AM-pole, which makes it difficult to correlate the localization between the PG and its synthase with segmented RT-qPCR used in our study, because the myometrium would be equally abundant in both segments. Further, the quantitative graph of PGF2a in the SI suggests that there are no significant differences of PGF2a between the WT and KO. For these reasons, we have chosen not to focus on this PG species, but still show its localization for future reference.

These points 4- and 5- are main concerns as title of the manuscript, subtitles of the Results and Discussion sections contain the general terms of PG localization, PG synthases, PG synthesis, PG biosynthesis....

Reply: We have revised the titles of the Results and Discussion. We have added a statement to the conclusion saying that despite only monitoring PGD2 in detail, these results can be interpolated to suggest similar discrepancies between other PG species and their synthases could occur. This is the first example showing quantitative comparisons between PG species and corresponding PG synthases, which shows a clear discrepancy in localization and abundance for the PG species we have studied. Additionally, mRNA for the rate limiting enzymes in PG synthesis Ptgs1 and Ptgs2 did not correlate with in-situ detection of PG abundance. We argue that showing synthase localization to be disparate with key PG species, in-situ measurements of PG species should be conducted to form meaningful biological hypothesis.

6- Panels showing COX2 FISH (Figures 2B and 3B) should be enlarged.

Reply: The panels have been enlarged

7- There are many inaccuracies in the numbering and the legends of figures.

Page 8-9: the text refers to Figure 3 instead of Figure 2.

Figure S1 has an incomplete legend.

Reply: We thank the reviewer for pointing out these mistakes and have amended them.

8- Figure S1: these results are only partially described in the Result section of the manuscript (Fig S1 is referred on page 5 but not on page 7).

The X axis should be modified according to that of Figure 2A panels (AM/M poles).

Why authors used another combination of synthases (Ptges, Ptges3 and not H-Pgds) and analyzed PGD2 receptor expression?

Some of the panels are redundant with panels of other figures (ex: Ptgds on Fig S1 and Fig. 2A/ Fig. 3A).

Reply: We thank the reviewer for pointing out that some descriptions for S1 were missing and have amended the manuscript accordingly. When updating the figures 2 and 3, the data from the previous figure S1 has been included in the main manuscript instead and the redundancies are removed. The X-axis have be modified for consistency between figures. We have removed the data for Ptges and Ptges3 since we did not detect any PGE2 and the inclusion of these data may cause confusion to the reviewers and readers as well.

9- The term qPCR is used all along the manuscript instead of RT-qPCR.

Reply: The term has been changed to RT-qPCR throughout in the entire manuscript

10- Authors should discuss why only PGD2 and PGF2a were detected using nano-DESI while Ptges is highly expressed at the AM pole, at a similar level to that of both Ptgds.

Reply: We have added a short description of this into the first results section stating that with the exception of PGD2 and PGF2a all other PGs were below our limit of detection, despite the likelihood of them being present in the tissue. In addition, we have clarified that although PGD2 and PGE2 are isomeric species, LC-IMS-MS showed that the vast majority of the detected species is PGD2 and any potential contribution by PGE2 is below the limit of detection.

11- Another main concern in this manuscript, is the lack of information on statistical analyses (authors should refer to the authors guidelines of the Communications Biology journal).

“The Methods must include a statistics and reproducibility section with the following information: the name of the statistical test, the n value for each statistical analysis, the comparisons of interest, a justification for the use of that test... Authors must state whether a number that follows the ± sign is a standard error (s.e.m.) or a standard deviation (s.d.).

Legends of Figures 2 and 3 do contain some informations; however, the statistical unit is not defined. Do three biological replicates mean three embryos from the same litter or from 3 different litters?

Reply: We thank the reviewer for this comment. Due to the different styles of data collected by RT-qPCR and mass spectrometry imaging we initially thought it would be more clear to include the proper statistical information in each respective section in the experimental section. The names of the statistical tests have been added to the experimental section of the manuscript. In addition, to Figures 2 and 3 captions we have added that the error bars represent 1 standard deviation and the statistical test performed. We have also added a section to the specified by Communications biology titled "Statistics and reproducibility."

The biological replicates mean three implantation sites from 3 different litters.

Reviewer #2 (Remarks to the Author):

Duncan et al. used a novel mass spectrometry imaging method to assess the localization and concentration of prostaglandins (PGs) in mouse tissues during pregnancy. Although the study is descriptive, this new method, a quantitative nano-DESI MSI, has clear advantages over whole tissue MS/MS analysis. This study should be of interest to Communications Biology's readers. It brings new biological insights on the spatial evaluation of PGs in tissues, which could reveal the functional significance of each PGs during different stages of embryo development/pregnancy. Although this work allows advancement in the field, there are several concerns that have to be addressed, as listed below.

Major concerns:

1. As this is a new method, the authors need to include a negative control (to control for a false-positive result). To strengthen the findings, the authors need to include a negative control in Figure 2, i.e., uterine tissues from non-pregnant mice.

*Reply: The uterine tissue is enlarged and differentiated as a result of embryo implantation. Compared to the ~ 4 mm in diameter uterine tissue on day 8 of pregnancy, the uterine tissue of non-pregnant mice does not contain the same morphological regions that are formed as a response to embryo implantation. In our previous publication, we showed localization of PGs to the luminal epithelium and the glandular epithelium using silver doped nano-DESI MSI at day 4 of pregnancy (Duncan et al. Quantitative Mass Spectrometry Imaging of Prostaglandins as Silver Ion Adducts with Nanospray Desorption Electrospray Ionization. Anal. Chem. **90**, 7246–7252 (2018)). However, these cellular regions no longer exist in the remodeled tissue on day 8 of pregnancy as clearly shown in the optical and ion images. Thus, the addition of tissue from non-pregnant mice would not work as a control in this case.*

2. The authors need to provide the clear goal of the study, to visualize PGD2 or to provide a new method? Or both?

Reply: We have added a statement at the end of the introduction to clarify to goal of this study.

"The goal of this study was to: i) visualize detectable PG species; ii) detect any perturbations to PG synthesis as a result of p53 deletion; and iii) correlate spatial distributions of PG species and PG synthases."

3. The authors showed that Ptg2 mRNA levels were significantly higher in Trp53d/d in the AM,

whereas the levels were lower in the M compared to Trp53f/f (Fig. 3A). However, it is difficult to see this result in the expression in the tissues in Figure 3B. To this reviewer, levels of COX2 in Figure 3B are similar between Trp53d/d and Trp53f/f tissues. Please provide larger images or zoomed-in images as insets.

Reply: The mRNA levels that are provided are quantitative measures and provide a good comparison between the Trp53d/d and Trp53f/f mice. The images showing FISH signals are not quantitative and should not be used to compare the levels of COX2 in the tissue. However, the FISH images show site- and cell-specific localization and have been enlarged in the manuscript according to the reviewer comment to ensure a better view of the localization. It is also important to note that these are included as reference without the statistical evaluations performed on the quantitative data from RT-PCR and MSI. The development of Trp53^{d/d} implantation sites show biological variations, since not all Trp53^{d/d} mice show compromised pregnancy outcomes.

4. Please use correct nomenclature throughout the manuscript, including the text and the figures, i.e., Ptgs1 and Ptgs2, italicized for mRNA and all capitalized, non-italicized for proteins. It is confusing at times when trying to interpret the results in Figures 2 and 3.

Reply: We thank the reviewer for pointing this out and have altered the text and figures accordingly. We have also rearranged the Figures 2 and 3 for clarity.

5. The explanation for the significance of anti-mesometrium vs. mesometrium needs to be introduced in the introduction, instead of the discussion (page 8, lines 14-19). This is important as the readers might not be familiar with the specifics of these two regions. In addition, not much of an explanation was included for the results in Figure 1.

Reply: In accordance with the reviewer suggestion, the explanation of the AM and M poles has been moved for an earlier introduction for the reader. In addition, the location of the AM and M poles has been added to Figures 1C and 1D. The result section now includes more details and results from the workflow showed in Figure 1.

6. Statement on page 8, lines 38-43, if that is the case, the authors need to include the measurement of PGD2 level in Ptgs2^{-/-} uterus. The PGD2 levels should be comparable between Ptgs2^{-/-} and Ptgs2^{+/+} uteri.

Reply: We agree with the reviewer that this would be a great comparison. However, it is impossible since Ptgs2^{-/-} females are infertile and therefore there will be no embryo implantation sites to study the PGD2 levels in. We have added the infertility of Ptgs2^{-/-} females into this statement to highlight the hypothesis.

7. Include the justification for the assessment of tissues from Trp53d/d mice on day 8 of pregnancy. This could confuse the readers as the authors mentioned on page 4, line 11 that 'deletion of p53 is elevated levels of COX2 activity on day 16 of pregnancy'.

Reply: Day 8 of pregnancy is the earliest day when the decidua is fully differentiated into primary and secondary decidual zones with polyploidy cells primarily in the secondary zone of WT mice. PGs plays a key role in the tissue remodeling around day 8. In Trp53^{d/d} mice, increased numbers of polyploid cells undergo terminal differentiation with the emergence of decidual cell senescence and retarded

development of the embryo (PMID: 20124728). The synthetic pathway of PGs on day 16 of pregnancy is clear, where PGs are produced by senescence-induced Cox2 expression. However, cox1 and cox2 are both expressed in day 8 implantation sites and Cox2 has been thought to be the major enzyme to produce PGs on day 8. However, the PGs showed different pattern as compared to that of Cox2. In late stage of pregnancy, deletion of P53 causes premature decidual senescence with increased levels of COX2; in the early pregnancy, our current study shows that Ptgs2 levels are increased at the antimesometrial pole of Trp53^{d/d} mice but decreased in Trp53^{d/d} mesometrial poles.

8. Please include the type of statistical analyses for all tests in the manuscript.

Reply: The types of statistical analyses has been added for all tests.

9. Please briefly explain how AM & M tissues were isolated. Were there any transcripts (markers) were measured for the purity of the isolation?

Reply: The following text has been added to the manuscript to describe this isolation: "Mesometrial and antimesometrial tissues were separated by the top edge of implantation chambers. The whole implantation site was separated into two halves along the M-AM axis, and the implantation chamber with an embryo was exposed. M and AM poles were obtained by cutting a half of the implantation site following a cutting plane vertical to M-AM axis at the top edge (close to the M pole) of the implantation chamber. The embryonic tissues were removed from the AM pole prior to..."

10. Please label "M" and "AM" on images in Figures 2C and 3C

Reply: The images have been labeled.

11. Please plot the raw value for each individual data point (in addition to mean) for Figures 2D and 3D.

Reply: The raw values have been added in Table S5 and S6.

12. Please consider presenting the data from M before AM as the tissues orient from top to bottom. This comment applies to Figures 3A and 3D. (switching data from M to the top and AM to the bottom panel)

Reply: The data has been switched in the figure to make it more clear to the reader.

Minor comments:

1. Some of "2" in PGD2 were subscripts, sometimes there were not. Please be consistent throughout the manuscript.

Reply: All have been changed to subscripts for consistency.

Reviewer #3 (Remarks to the Author):

The authors used silver-doped nanospray desorption electrospray ionization (nano-DESI) MSI to reveal in situ localization and abundance of prostaglandin species along with situ localization of prostaglandin synthetic enzymes in mouse embryo implantation sites on day 8 of pregnancy. They

found PGD2, its precursors, and downstream synthases co-localize with the highest expression of COX1, but not COX2. Further, they found that the abundance of COX and prostaglandin D2 synthases in cellular regions does not mirror the regional abundance of prostaglandins when conditional deletion of transformation-related protein 53, a known regulator of COX2-derived prostaglandin signaling, was carried out. They deduce that prostaglandins tissue localization and abundance may not be inferred by COX or prostaglandin synthases in uterine tissue, and must be resolved by an in situ prostaglandin imaging. Their findings, particularly the part on PGD2, are interesting and may be of interest to reproductive biologists to study role of PGD2 in implantation and early development. However, there are some concerns which requires the authors' attention.

Major:

1. It may be true that mere measurements of PG synthetic enzymes may not mirror the abundance of a particular PG specie. However, we should also bear in mind that PGs are quite diffusible and their abundance at a particular site is determined not only by their synthesis but also by their diffusion and degradation. I suggest the authors consider the diffusion factor and look at the enzyme responsible for their degradation at the same time.

Reply: We agree with the reviewer that these points are of utmost importance for understanding PG localizations. We show in Figure 3c that Pdgh levels are consistent in the AM and M poles, thus enzymatic degradation is not the reason for the lower abundance of PGD2 in the M pole compared to the AM pole. Further, it is not reasonable to assume that spontaneous degradation of PGD2 should be significantly different between the AM pole and the M pole of the same tissue and give rise to these very distinct localizations. Therefore, we rule out degradation as a reason for the different abundances in the two poles. Regarding diffusion of PG between regions, we agree that there may be some diffusion. However, we argue that concentration gradient driven diffusion of individual PG species from the M pole to the AM pole (~ 2mm) could not explain the significant differences detected between the regions. Moreover, the well-defined and repeatable shapes of PG species localization suggest that free diffusion is not substantial. We cannot find any evidence in the literature that would suggest PG specific transport from one tissue region to another. Moreover, the PG species co-localize with PG precursors such as AA and 2-AG. As a result, we maintain that diffusion may contribute to delocalization throughout the tissue, but not at the scale observed in mass spectrometry imaging experiments. Further, our conclusion that in-situ detection of PG species in tissue hold true regardless, as PG synthase abundances and localizations are disparate from PG species in tissue.

2. It is well known that a number of other PGs are involved in implantation as well. The authors focused mainly on PGD2 and PGF2alpha in this study. I wonder whether the claimed phenomena also hold true in other PG species.

Reply: The focus on PGD2 and PGF2alpha in this study is not because of the lack of interest in other PG species, in fact we have looked for all other PG species in our data sets. However, with this new technology the only PG species above our limit of detection and limit of quantification are PGD2 and PGF2alpha. We hope that we can further improve our technique to enable quantitative imaging of additional species at lower abundances in the future.

3. Mostly, the authors used mRNA abundance to reflect the abundance of PG synthetic enzymes. Apparently, mRNA level may not necessarily reflect protein level. Is it possible to measure the protein level at the same time?

Reply: We agree that mRNA levels does not necessarily reflect protein level. However, with the low protein abundances in these small tissue regions it is hard to measure protein levels. Some successes

of protein distributions in smaller regions of tissue sections have been reported in this regard, for example using nano-POTs (Zhu, Y., Piehowski, P.D., Zhao, R. et al. Nanodroplet processing platform for deep and quantitative proteome profiling of 10–100 mammalian cells. Nat Commun 9, 882 (2018)), but additional technological advancements are needed to fully realize the correlation between mRNA levels and protein abundances.

Minor:

1. For the abstract, the authors should introduce the background on p53 in terms of PG synthesis. Otherwise, it comes out of blue and is hard to understand.

Reply: We have added a short background on p53 in pregnancy and PG relevance to the abstract.

2. Because the methodology is also an important part of the manuscript, I would suggest that the authors put all supplementary figures into the manuscript.

Reply: We are hesitant to follow this suggestion by the reviewer because we fear that the message of PG metabolites and synthases localizations might be diluted and lost. Therefore, and unless the editor persists, we will keep most of the figures in the supporting information and have only moved the RT-qPCR data from figure S1 to figure 3.

3. Can you show the data that only PGD2 and PGF2 α were detected by MSI ?

Reply: We acknowledge that this is an important point, however, we don't see the reason for adding empty mass spectra or blank ion images into the manuscript. Instead we amended the text to clearly state that these are the only two PG species that were detected in our imaging data sets.

REVIEWERS' COMMENTS:

Reviewer #1 (Remarks to the Author):

This manuscript is much improved.

The authors have addressed my concerns in the Results, Discussion and Materials and methods sections as well as in the figures. They also developed and reorganized the Introduction section.

However, I still feel the Figure 3 needs to be modified: Figures 3B and 3C that contain redundant information, might be grouped.

Reviewer #2 (Remarks to the Author):

The authors have adequately addressed all concerns.

Reviewer #3 (Remarks to the Author):

The authors have addressed all my concerns.

REPLY TO REVIEWERS' COMMENTS:

Reviewer #1 (Remarks to the Author):

This manuscript is much improved.

The authors have addressed my concerns in the Results, Discussion and Materials and methods sections as well as in the figures. They also developed and reorganized the Introduction section.

However, I still feel the Figure 3 needs to be modified: Figures 3B and 3C that contain redundant information, might be grouped.

Reply: We fear that combining Figures 3B and 3C may overload the data presentation, limit the readers' ability to decipher the presented data, and discourage the readers to pay closer attention to results. Thus we have not modified the figure as suggested by the reviewer.

Reviewer #2 (Remarks to the Author):

The authors have adequately addressed all concerns.

Reviewer #3 (Remarks to the Author):

The authors have addressed all my concerns.